# A Machine-Learning Approach to Measure the Anterior Cruciate Ligament Injury Risk in Female Basketball Players

**DOI:** 10.3390/s21093141

**Published:** 2021-04-30

**Authors:** Juri Taborri, Luca Molinaro, Adriano Santospagnuolo, Mario Vetrano, Maria Chiara Vulpiani, Stefano Rossi

**Affiliations:** 1Department of Economics, Engineering, Society and Business Organization (DEIM), University of Tuscia, 01100 Viterbo, Italy; luca.molinaro@unitus.it (L.M.); stefano.rossi@unitus.it (S.R.); 2Motustech—Sport & Health Technology c/o Marilab, Ostia Lido, 00012 Rome, Italy; 3Physical Medicine and Rehabilitation Unit, Sant‘Andrea Hospital, “Sapienza” University of Rome, 00189 Rome, Italy; adrianosantospagnuolo1@hotmail.it (A.S.); mario.vetrano@uniroma1.it (M.V.); mariachiara.vulpiani@gmail.com (M.C.V.); 4Sports Medicine Institute CONI Rome, 00197 Rome, Italy

**Keywords:** machine learning, inertial sensors, basketball, ACL injury, leg stability, leg mobility, load absorption, Landing Error Scoring System

## Abstract

Anterior cruciate ligament (ACL) injury represents one of the main disorders affecting players, especially in contact sports. Even though several approaches based on artificial intelligence have been developed to allow the quantification of ACL injury risk, their applicability in training sessions compared with the clinical scale is still an open question. We proposed a machine-learning approach to accomplish this purpose. Thirty-nine female basketball players were enrolled in the study. Leg stability, leg mobility and capability to absorb the load after jump were evaluated through inertial sensors and optoelectronic bars. The risk level of athletes was computed by the Landing Error Score System (LESS). A comparative analysis among nine classifiers was performed by assessing the accuracy, F1-score and goodness. Five out nine examined classifiers reached optimum performance, with the linear support vector machine achieving an accuracy and F1-score of 96 and 95%, respectively. The feature importance was computed, allowing us to promote the ellipse area, parameters related to the load absorption and the leg mobility as the most useful features for the prediction of anterior cruciate ligament injury risk. In addition, the ellipse area showed a strong correlation with the LESS score. The results open the possibility to use such a methodology for predicting ACL injury.

## 1. Introduction

Basketball is one of the most widespread team sports with more than 450 million amateur and professional players in the world [1]. During both training sessions and competitions, athletes are asked to perform dynamic movements, requiring also physical contact between players. Moreover, basketball is a vertical sport, which requires jumping and landing activities two or three times more often than other team games, such as soccer and volleyball [2]. These aspects lead to a high incidence of injuries among basketball players; specifically, the knee joint has been demonstrated as the most commonly stressed and injured body area [3]. Among others, anterior cruciate ligament (ACL) rupture can be considered as the most debilitating injury, often leading to extended rest periods before the return to play [4]. In addition, ACL injury, together with ankle sprains, has the main incidence in female basketball players; in fact, 16% of them may incur an ACL injury during their career. It is worth highlighting, as the risk of ACL rupture in female players is up to eight times more than male players [5]. The treatment of ACL injury always requires reconstructive surgery, which leads to the absence from playing field for at least six months, and a rehabilitation program that may be not sufficient to recover the complete functionality of the knee and the athlete’s skills before the injury [2]. Several studies have been conducted to understand the main underlying risk causes, as the combination of anatomical, physiological and biomechanical factors can cause an increase in the injury occurrence [6,7,8].

From this perspective, different methodologies have already been validated to provide useful information for the identification of athletes at higher risk, aiming at proposing dedicated programs that can reduce the injury incidence [9,10,11]. In this context, optoelectronic systems combined with force platforms still represent the gold standard in order to have accurate and repeatable measures of selected movements, such as crossover test, pivoting jumps and countermovement jumps [12,13,14]. In addition, video analysis has shown its potential in the identification of ACL injury risk by offering the possibility to record data during sport activities with low-cost devices and easy post-processing operations that do not require skilled operators [15,16]. From the video analysis approach, a predictive methodology for knee injury has been implemented and validated, named the Landing Error Scoring System (LESS) [17,18]. The LESS is a clinical screening tool able to identify potential ACL injury, consisting of the analysis of biomechanical factors based on three jumping tasks [18]. However, LESS has shown several limitations and its validity is still questioned, mainly due to the subjectivity of the video analysis that leads to a poor inter-operator reliability of the procedure [19,20,21]. In the last decades, wearable solutions have been proposed for overcoming the intrinsic limitation of the optoelectronic systems. In fact, it is well known that the optoelectronic systems cannot be used to perform an in-field experimental protocol if the light conditions are not constantly controlled; in addition, wearable solutions often allow reductions in instrument costs [22,23].

Taking into account the above-reported existing methodologies and their limitations, as well the recent advances in artificial intelligence (AI), the application of machine-learning algorithms appears to be a promising approach for diagnosis and prediction in several fields, such as ACL injury [24,25,26,27,28,29,30], and, more generally, in biomedicine approaches [31,32,33]. Focusing on ACL, diagnosis and prediction represent two correlated analyses that permit us to solve two different issues. In fact, an early diagnosis allows both a reduction in incorrect evaluation at initial presentation, which has been shown to be up to 14% of cases [34], and the minimization of the delay to surgery. Conversely, an early prediction leads to a reduction in the probability of suffering from the effective ACL rupture by adopting specific training programs [26]. Considering diagnostic aspects, Lao et al. provided an overview on the application of AI on magnetic resonance imaging (MRI) able to detect the ACL injury [27], among others. The study proposed by Mazlan et al. showed the support vector machines were able to classify up to 100% of three different ACL injuries, which were normal, partial and crucial [24]. Similar results were obtained by applying a convolutional neural network on coronal MRI, reaching an accuracy of 96% in ACL diagnoses [25]. By moving to the use of AI as a predictive tool, Jauhieinen and colleagues demonstrated a random forest was able to detect the most predictive biomechanical factors, which were mostly related to the knee joint kinematics and kinetics, during vertical jumping tests performed by 314 basketball and football players [28]. They also assessed the difficulty in the prediction of future injuries, also confirmed by the low value of the area under receiver operator curve, lower than 0.70 for both random forest and logistic regression. Similar accuracy results were obtained by Oliver et al., who tested 355 football players during single-leg countermovement jump and single-leg hop for distance tests, analyzing data related to the knee kinematics [29]. Moreover, Tedesco et al. demonstrated that a gradient boosting algorithm, fed with data related to the acceleration and stability of the leg, was able to discriminate, with an accuracy close to 82%, between healthy players and post-ACL injury subjects. Thus, this approach can be useful for the identification of the return to sport time as a monitoring tool [30]. 

Although it is evident the promising usefulness of artificial intelligence for providing automatic information to both diagnose and predict ACL injury, the obtained results are not completely satisfactory to propose the use of such methodologies in training programs. In fact, the already published findings are often characterized by a greater percentage of false negative occurrences, which are severe misclassifications since they lead us to consider athletes at no risk at higher risk. In addition, to the best of the authors’ knowledges, no studies have evaluated neither the correlation between the use of the AI approach and the predictive tool actually used, such as the LESS score, nor the feasibility of using data related to the leg stability and load absorption capability to feed machine-learning algorithms. Thus, this paper aims at understanding the feasibility of machine-learning algorithms to classify basketball players at higher ACL injury risk by using the LESS score as a supervisor for algorithm training. More specifically, we conducted an observational study on female basketball players when performing monopodalic vertical jump and single-leg squat tests and compared different machine-learning algorithms fed with features extracted from inertial data concerning leg stability, leg mobility and load absorption capability. The results of this study could offer the possibility to use AI for the automatic identification of athletes at higher risk, allowing us to implement customized training programs addressed to injury prevention. 

## 2. Materials and Methods

### 2.1. Participants

Thirty-nine basketball players belonging to the female Under 14 Italian team (height = 170.5 ± 6.3 cm, body mass 62.9 ± 4.9 kg, age = 13 ± 1 years) were involved in the experimental protocol. Athletes were included in the study if: (i) they did not suffer from severe injuries in the last three years, (ii) they trained at least five times per week and (iii) their parents, or tutor, accepted the written informed consent. Any injuries at knee level, both severe and moderate, caused the exclusion from the test. All the tests in the experimental protocol were decided with the coach and athletic trainer and they met the criteria required by the Declaration of Helsinki. 

### 2.2. Experimental Setup and Protocol

The experimental setup consisted of three sensor systems: an inertial measurement unit—IMU (GyKo, Microgate S.r.l., Bolzano, Italy), optoelectronic bars (Optogait, Microgate S.r.I, Italy, 2010) and cameras. The IMU is composed of 3D linear accelerometers, gyroscopes and magnetometers, allowing us to measure linear acceleration, angular velocity and magnetic field vector, with a full scale of ±16 g, ±2000 °/s and ±4800 μT, respectively. The sensor was placed on the shank using an elastic belt, appropriately designed to reduce the movement artifacts. Four optoelectronic bars were used in order to create a square field of dimensions 2 × 2 m^2^; each bar consisted of 96 LED diodes, which allowed us to estimate both the flight and contact time during jumps. Finally, two cameras (Logitech C920 HD) were used to acquire both sagittal and frontal plane images and placed at 2 m from the participant. The experimental setup is shown in Figure 1. 

The experimental protocol consisted of three phases: warm-up, clinical assessment and sensor-based assessment.

With regard to the first phase, a 10-min warm-up for the lower limb joint was conducted, including standard stretching exercises for lower limb mobility and five minutes of biking. 

Successively, the clinical assessment phase was performed in order to compute the LESS score. Each athlete was asked to perform a jump-landing task that involved both vertical and horizontal movements; specifically, they jumped forward from a 30 cm box and after the landing they immediately performed a maximum vertical jump. The jump-landing task was repeated three times. During this phase data were gathered from the two cameras. 

Finally, the sensor-based assessment was composed of two motion tasks. Firstly, each participant was asked to perform five consecutive monopodalic countermovement jumps (mCMJ) [35], receiving the instruction to complete this movement sequence (Figure 2a): (i) reach a single-leg starting position with the foot shoulder-width apart; (ii) perform a downward movement; (iii) immediately execute a concentric phase; and (iv) repeat point (ii) and (iii) five times. During all movements the arms were free to move. The second task consisted of a single-leg squat (SLS), in which the participant had to perform a squat on a single leg maintaining the arm with the hands on the hip and extending the other leg in front of the body [36], as in Figure 2b. Five repetitions of SLS were performed. Both GyKo and optoelectronic bars were used during the first task, whereas only the GyKo was used for gathering data during the second one. 

All the tests were performed with the dominant leg, which was selected as the one used to kick a ball [37]. The entire protocol lasted 20 min per athlete, also considering the time for the sensitization and the time needed for a static acquisition in upright position before each motion task.

### 2.3. Data Analysis and Feature Extraction

#### 2.3.1. Clinical Assessment Phase: LESS Score

Data acquired during the jump-landing task were analyzed through a video analysis performed by a skilled operator. The feet, ankle, knee, hip, trunk, shoulder, neck and head posture was assessed through the video and the LESS score was obtained by analyzing the landing technique error based on a 17-items questionnaire shown in the Table 1, as proposed by Padua et al. [18].

In more detail, items from 1 to 6 indicate the jump-landing quality in terms of lower limbs and trunk positioning at the initial contact; from 7 to 11 the errors in positioning of the feet; from 12 to 14 assessing the lower extremity and trunk movements between initial contact and the moment of maximum knee flexion angle; the item 15 at the moment of maximum knee valgus; and finally the last two items indicate the overall quality of the motion gesture. A maximal score of 19 can be achieved, representing the worst jump-landing performance. 

It has been demonstrated that a LESS score value equal to or lower than 5 is associated with athletes who can be considered not at risk of ACL injury; conversely, a score greater than 5 indicates athletes at risk [38]. The evaluation was performed for each of the three jumps; then, the median value was considered to define athletes at risk of injury and used as a reference for the further analyses. 

#### 2.3.2. Sensor-Based Assessment Phase: mCMJ and SLS

Considering both tasks, the data acquired during the static calibration in upright position were firstly used to perform the re-alignment of the sensor axes with the absolute reference system; then, the Mahony filter was applied to extract the orientation of the sensors by combining linear accelerations and angular velocities [39]. 

Two sets of parameters were computed for assessing the leg stability and the load absorption capability. 

##### Leg Stability

Linear accelerations and angular velocities were used to estimate the orientation of the vertical axis ($\vec{S}$) of the sensor. The path covered by the projection of $\vec{S}$ on the horizontal plane placed at 1 cm below the inertial sensors was analyzed with parameters typically used in posturography. Following the equations proposed by Prieto et al. [40], we computed the following stability parameters:The total length of the path in the plane, named PL;The total length of the path in antero-posterior direction, named PL_AP_;The total length of the path in medio-lateral direction, named PL_ML_;The area of the bivariate confidence ellipse that includes at least 99% of the $\vec{S}$ projection points, named EA.

These parameters were computed for both tasks. For mCMJ, the parameters were then normalized for the stabilization time (Ts), which was defined as the time between the initial contact time with the ground, detected by the on status of the optoelectronic LED diodes, and the instant in which the leg could be considered stable, which was associated with the first minimum of the absolute value of the shank’s angular velocity after the contact time, which guarantees to take into account the time period identified as the most likely for injury occurrence [41]. Conversely, for SLS the parameters were normalized for the duration of the descending phase (T_DP_) since it is associated with a greater stress for the knee joint with respect to the ascending phase [36]. T_DP_ was defined as the time between the start and the end of the movement. Specifically, the start of movement was identified as the instant in which the standard deviation of the module of the three angles overcame five times the value of the average SD obtained during the static trial; the end of the movement was identified as the maximum value reached by the angle around the y axis, which approximately corresponds to the knee flexion axis, named θ_ymax_. The θ_ymax_ was considered as the index related to the leg mobility; specifically, the lower the value, the lower the mobility [42,43].

The leg stability parameters allowed us to quantify the leg stability; in fact, it was shown that greater values corresponded to greater instability after the jump [22], which is considered as one of the main causes of ACL injury [41].

##### Load Absorption Capability

Load absorption capability was assessed only related to mCMJ. The root mean square (RMS) of the shank linear acceleration was computed both considering the vertical component (RMS_z_) and the module of the components along the xy horizontal plane (RMS_xy_), both expressed in m/s^2^. The parameters were normalized to Ts. Greater values of RMS indicated the poor capability of the athlete to absorb the load during the landing phase [44]. 

All the examined parameters, summarized in Table 2, were considered as features for the training and testing of the machine-learning algorithms in order to distinguish between two classes: athlete at risk or not, hereinafter class R and class NR. Regardless of the type of parameter, all of them were averaged, finally, across the five jumps or the five squats.

### 2.4. Machine-Learning Algorithms

In this study, we decided to examine two different categories of supervised machine-learning algorithms, i.e., geometric and binary, since they represent two of the most widespread algorithms for motion recognition [45]. Among geometric classifiers, we selected the support vector machine (SVM) and the k-nearest neighbor (kNN); instead, the decision tree (DT) was chosen among binary algorithms. 

*SVMs* are geometric supervised machine-learning algorithms that work on the identification of the hyperplane that guarantees the best separation of the features related to different classes. The kernel function, which is used to linearize the feature space, represents the fundamental parameter to be selected before the classification process. Specifically, we here tested three SVMs, considering the kernel function as linear (l-SVM), quadratic (q-SVM) and cubic (c-SVM). 

*kNN* are geometric supervised machine-learning algorithms that perform the classification decision by identifying the most common class among the k-nearest neighbors and maximizing the distance among other classes. Thus, the equation for the computation of the distance represents the first parameter to be selected. Specifically, we here tested three kNN: the fine kNN (f-kNN), which considered the Euclidian distance to recognize different classes with the number of neighbors set to 1; the cosine kNN (c-kNN), which considered the cosine distance and the number of neighbors set to 10; and the weighted kNN (w-kNN), which used the weighted Euclidian distance based on the squared inverse approach by selecting the number of neighbors equal to 10. 

*DTs* are binary supervised machine-learning algorithms that predict the most likely class by creating a set of nodes, in which the classification process is performed through specific split criteria. In this study, the split criterion was based on the Gini index. In addition, the maximum number of splits represents a fundamental parameter to be selected before the classification process. Specifically, we here tested three DTs: the coarse (c-DT), medium (m-DT) and complex DT (cx-DT), in which the maximum number of splits was equal to 4, 20 and 100, respectively. 

For further theoretical details on the selected classification algorithms, please refer to [46].

To summarize, we here compared nine machine-learning algorithms, as reported in Table 3. 

By using the above-mentioned features, we evaluated the performance of all the selected machine-learning algorithms with the application of a 9-fold cross-validation. In this way, 30 subjects comprised the dataset of training, and the remaining nine the dataset of validation, in turn. Cross-validation is an approach that guarantees the robustness of the obtained performance [46]. Then, the algorithm performance was assessed by averaging across all the folds. 

### 2.5. Performance Evaluation

The classes, risk and no risk, estimated by means of the above-mentioned algorithms were compared by the reference value obtained through the LESS score. For each classifier, a 2 × 2 confusion matrix was then obtained. Successively, the algorithm performances were measured in terms of accuracy, F1-score and goodness index [46]. 

Accuracy (*A*) was computed as the ratio between the correct predicted risk classes and the total of predictions though the following Equation (1):(1)$$A=\frac{TP+TN}{TP+TN+FP+FN}$$
where *TP*, *TN*, *FP* and *FN* represent true positive, true negative, false positive and false negative, respectively. Considering the positive as the class “no risk” (NR), a true positive and true negative were obtained when the algorithm correctly classified the athlete at no risk (NR) and at risk (R), as they were classified by the LESS score. A value close to 1 represents the perfect classifier, and 0.80 is usually selected as threshold value for considering an optimum classifier [46]. 

Even though the class imbalance in the present study can be considered slight, as in accordance with [47], we decided to compute additional metrics in order to avoid bias in the results due to the higher number of no risk athletes in the dataset [48]. Specifically, F1-score and G-index were computed.

The F1-score was computed as a harmonic average of the recall (*Re*) and the precision (*P*) value; where the recall represents the ratio between the true positive and the sum of the true and false positive, and the precision (*P*) was, instead, computed as the ratio between the true positive and the sum of the true positive and false positive. F1-score was calculated according to the following Equation (2):(2)$$F1-\mathrm{score}=2\cdot \frac{\left(Re\cdot P\right)}{\left(Re+P\right)}$$

As for the *A*, F1-score can range from zero to one, with a value close to 1 indicating a perfect classifier and a threshold value of 0.80 identifying an optimum classifier. 

Moreover, the goodness index (*G*) that represents the Euclidian distance in the receiver operating characteristic space between the tested classifier and the perfect one was computed as in Equation (3):(3)$$G=\sqrt{{\left(1-TP\right)}^{2}+{\left(1-TN\right)}^{2}}$$

*G* assumes values from 0 to $\sqrt{2}$ and the following goodness range can be considered: (i) optimum when *G* ≤ 0.25; (ii) good when 0.25 < *G* ≤ 0.70; (iii) random if *G* = 0.70; and (iv) bad if *G* > 0.70 [46].

After the validation of the algorithms, the feature importance was measured through the mean decrease accuracy (MDA) based on random forest [49,50]. MDA represents the average loss of accuracy achieved by the random forest on data out-of-bag (OOB), where the OOB approach consisted of three steps. Firstly, one feature in turn was excluded from the model and the prediction error was computed; secondly, the predictor variables of the new model were permuted to lose the correlation with the true class and the prediction error was computed on this new dataset; and, finally, the difference between the two predictor errors was computed and averaged across all the turns to obtain MDA, expressed as a percentage. The greater the value of MDA, the greater the feature importance.

Finally, the correlation between the most important features and the LESS score was computed by means of Spearman’s correlation tests, considering a significance level set to 0.05. The absolute value of r can be interpreted as: (i) no correlation, if |r| ≤ 0.1; (ii) mild/modest correlation, if 0.1 < |r| ≤ 0.3; (iii) moderate correlation, if 0.3 < |r| ≤ 0.6; (iv) strong correlation, if 0.6 < |r| < 1; and, finally, (v) perfect correlation, if |r| = 1, as in [37,51].

## 3. Results 

The results of the LESS analysis led to the identification of 26 athletes belonging to the class NR and the remaining 13 to the class R. In particular, the mean value of the LESS was equal to 3.5 and 7.5 for the NR and R group, respectively. As an example, the path length of the vertical axis projection on the horizontal plane obtained during a jump task, as well the waveform of the angle θ during a squat task performed by one subject belonging to either the R or NR group, are reported in Figure 3.

It is possible to observe how the subject belonging to the R group was characterized by a greater excursion of the path length in the ML direction, leading also to a greater value of the ellipse confidence area when landing after a monopodalic countermovement jump, as well as by a lower value of the knee angle during a single-leg squat movement. More generally, the mean values and standard deviations of the computed features for each group are reported in Table 4. By analyzing these results, it can be observed that athletes in the R group generally showed a greater path length in the ML direction and a greater value of the EA than those in NR, as well as a reduction in stabilization time related to the leg stability evaluation when performing mCMJ. Similarly, the load absorption parameters were found to be higher when computed on data gathered from athletes at risk of injury. Focusing on SLS, stability parameters showed similar behavior in both groups; conversely, the R group was characterized by a lower leg mobility and a lower descending phase time. 

Figure 4 shows the confusion matrices calculated for all the examined classifiers. 

Table 5 shows the results in terms of accuracy, F1-score and goodness index achieved by all the tested classifiers and derived from the confusion matrices. 

By focusing on the accuracy, the linear SVM achieved the best results, equal to 0.96; instead, the worst was associated with the cosine kNN with a value of 0.67. More generally, six out nine of the tested classifiers overcame the optimum threshold set to 0.80. Similar outcomes were obtained by considering the F1-score, whose values ranged from 0.71, which was associated with the cosine kNN, to 0.96, which was related to the linear SVM. With regard to the F1-score, the same six out nine classifiers achieved the performance of an optimum classifier. By considering the *G* index, the best performance was achieved by the l-SVM, i.e., a value equal to 0.08, and the worst by the c-kNN, i.e., a value equal to 0.46. Concerning *G*, only five classifiers fell into the optimum range with the cubic SVM that passed only the threshold related to the accuracy and F1-score. 

The MDA values for all the used features, in order to understand their importance, are reported in Figure 5.

By analyzing the histograms, the ellipse area computed during the monopodalic countermovement test was associated with the highest value of the MDA, i.e., 49.2%, leading us to define it as the most important feature of the dataset. A high importance was also found for the RMS_xy_ computed during the mCMJ; an MDA greater than 15% was also achieved by the RMS_z_ gathered during the mCMJ and the θ_ymax_ computing for the single-leg squat. MDA values lower than 5.0% were obtained by all the other features.

Considering the correlation analysis, the four features that reached an MDA at least equal to 15.0% showed different behavior, as reported in Table 6. 

Considering the goodness of the correlation, EA was found to be the only parameter that strongly correlated with the LESS, whereas RMS_xy_ and θ_ymax_ reached a moderate correlation, and RMS_z_ a modest correlation. 

## 4. Discussions

This paper presents an investigation on the feasibility of using a machine-learning approach to measure the risk of anterior cruciate ligament injury in female basketball players and to propose such a methodology in training programs. Through this scope, a comparison among nine machine-learning algorithms fed with data gathered from an inertial sensor and optoelectronic bars was carried out.

### Is It Possible to Use a Machine-Learning Approach to Measure the ACL Risk?

By analyzing the general results shown in Figure 3 and Table 4, it is confirmed that athletes at a higher risk of injury are typically characterized by a reduction in knee stability and mobility, well recognized as two of the main causes of ACL injuries [4,52]. In addition, the qualitative differences related to the two examined groups open the possibility to use such parameters as features to feed machine-learning algorithms. In fact, it is well known that a machine-learning approach is more promising when classes to be recognized are characterized by distinguishable motor patterns [46].

Such a possibility is strongly confirmed by the optimum performance achieved by the tested classifiers, especially when referring to the linear SVM. The selection of the right model parameters, such as kernel function for SVM, distance equation for kNN and number of splits for DT, appears to be a fundamental step for the classification process, since it can lead to a worsening of the algorithm performance. This result is in line with previous studies, in which it was demonstrated that the tuning of model parameters should be always conducted to avoid misclassifications [53,54]. After the right selection of the model parameters, by combining the results we can assess that l-SVM, q-SVM, f-kNN, m-DT and cx-DT meet all the criteria to be considered as optimum classifiers. Among the five optimum classifiers, the linear support vector machine reveals itself as the best, considering all the synthetic indices. In addition, we can assess that the l-SVM should be chosen not only for the highest value of *A* and F1-score, as well the lowest of *G*, but also taking into account that: (i) it is characterized by a higher value of prediction speed, a lower memory usage and an easier interpretability of the results than q-SVM due to the lower order of the kernel function [55]; (ii) SVM outperforms kNN in managing outlier data, reducing the number of false positives and false negatives [56]; and (iii) geometric classifiers are generally more robust across subjects than decision trees, which apply a threshold-based classification logic [57].

Furthermore, by considering the results related to the feature importance, we can speculate that the ellipse area, RMS_xy_ and RMS_z_ computed during the jump task, as well as the θ_ymax_ computed for the squat task, present significant differences that permit us to distinguish the athletes at higher risk ACL injury. Consequently, the involvement of the remaining nine parameters among the feature set can be avoided, permitting us to reduce the computational time of the automatic classification algorithm. The sensitivity of the parameters related to the load absorption capability and the leg mobility in the recognition of athletes at higher risk is, instead, in line with the outcomes already reported in the literature. Concerning the load absorption, Shimokochi et al. [58] demonstrated that prevention programs addressed to increasing the athlete ability in reacting to the load during the landing phase after a jump can help to reduce the risk of injury. In addition, this aspect is emphasized in female athletes, who adopt an absorption strategy that consists of the use of distal joints, causing an increase in the risk of ACL injury with respect to male players [59]. As for the leg mobility, knee angle joint has been shown as one of the main aspects to be investigated during motion tasks for the assessment of ACL injury risk in different sports, such as basketball, soccer and volleyball [16,60]. Among the stability indices, only the confidence ellipse area reveals itself as a useful predictor in this context; this finding is in contrast with those reported in the study conducted by [61], in which the absence of a correlation between knee stability parameters and ACL injury was assessed in soccer and handball players. However, it is worth noting that the outcomes in [61] were obtained through semi-static tests, i.e., the execution of the star excursion balance, rather than jump-based tasks, and with a different population; indeed, it is recognized that the evaluation of the instability during the landing phase after a jump can be used as a predictor factor [41]. By combining the results on the best performing classifier and the most important feature, we can assess that the evaluation of leg stability, leg mobility and load absorption capability in athletes performing monopodalic jump and single-leg squat tasks and the successive application of a machine-learning algorithm based on a linear support vector machine can be a useful screening tool for the identification of athletes at higher ACL injury risk. In addition, it is also possible to suggest performing only the monopodalic countermovement jump, since the exclusion of the θ_ymax_, which is the only important parameter related to the squat task, did not lead to a significative change in the algorithm performance. One of the main advantages of the proposed algorithm can be ascribed to the possibility to apply such a procedure without the necessity of a skilled operator, which is fundamental for the computation of the LESS score in order to avoid the limitation related to the subjectivity of the operator [20].

### Implications

Considering our findings, we can speculate that the linear SVM, moreso than others, may be introduced as a screening tool for the identification of the athletes at a higher risk of injury if it is fed with data gathered from athletes when performing jumps and squat tasks. In addition, the use of artificial intelligence based on quantitative and objective measures could permit us to overcome the intrinsic limitation of a subjective evaluation, which the landing error scoring system is. The performance achieved here by the test classifiers are encouraging results, since they are up to more than 20% higher with respect to the previously published applications of artificial intelligence in this context [29,30]. This improvement can be ascribed to a greater robustness of the linear SVM to the type II error; in fact, in the previously published studies the limitation was the high value of false negatives, as in [29]. Since the cited papers only used ground reaction force during the landing phase, we can speculate that the computation of additional parameters, such as the stability ones and those related to the linear acceleration, leads to the reduction in the false negative classification. 

Finally, the strong correlation found for EA permits an objective evaluation of the knee stability, revealing itself as a predictive factor, and a better quantification of changes over time also when functional scales appear no longer sensitive. In particular, since EA index is not a discrete variable, unlike the functional scales used in clinics, it could permit a greater resolution in the discrimination of different risk levels; as well, by not presenting a maximum possible value, it could guarantee to overcome the well-known intrinsic limit of clinical score associated with the saturation effect at the upper limit of the scale [62]. These considerations are still true if considering other clinical scales, such as the Cutting Movement Assessment Score (CMAS), a validated qualitative screening tool to identify athletes at higher risk [63]; though it is worth noticing that CMAS is specifically addressed to sports in which side-step cutting is a predominant mechanic, such as rugby and soccer [64]. 

By summarizing, we can affirm that the method proposed here could be positively exploited during training sessions since it only requires: (i) the execution of easy motor tasks typical of training programs, since their correct execution is shown to be correlate to the prevention of ACL injury [65,66] and (ii) the use of only one low-cost wearable inertial sensor, which has already been demonstrated to be useful for the assessment of ACL injury [67]. Using this protocol as a screening method, customized interventions could be implemented for the athletes identified as potentially at higher risk of injury. Thus, we can conclude that practitioners are encouraged to include the protocol and data analysis methodology validated here during the training sessions to constantly monitor athletes at higher risk. In addition, it has to be noted that such an approach can be also used to monitor the effectiveness of customized interventions, which are tailored training programs, for the reduction in the injury risk.

Even though this study is promising, it should be highlighted that the obtained results are only related to the tested cohort. Thus, further studies should be carried out in order to understand if such a methodology can be extended to both other contact sports and populations with different demographic characteristics, such as age and gender.

## 5. Conclusions

With the aim to assess the feasibility of using a machine-learning algorithm for the identification of basketball players associated with a higher risk of anterior cruciate ligament injury, we compared nine different classifiers fed with data related to leg stability, leg mobility and load absorption capability. Data were gathered from thirty-nine female basketball players when performing single-leg countermovement jumps and single-leg squats. The Landing Error Scoring System was used to identify the athletes at higher risk of injury and as a supervisor for the machine-learning algorithms. The results reveal that a support vector machine implemented with linear kernel achieved the best performance in terms of accuracy, F1-score and goodness index, equal to 0.96, 0.95 and 0.06, respectively. By analyzing the feature importance, the ellipse area and root mean square of the acceleration measured during the jump test, as well as the angle reached during the single-leg squat, we showed the greatest sensitivity to discriminate if an athlete is at risk of injury. In addition, the ellipse area strongly correlated with the LESS score. The obtained findings permit us to validate the use of the artificial intelligence approach as a predictive tool for anterior cruciate ligament injury monitoring. Thus, the proposed methodology should be adopted by practitioners during training programs to monitor athletes at higher risk and, eventually, implement customized programs to reduce the risk level. Future works will consider the application of such a methodology on players of different contact sports, such as soccer or handball, and also taking into account the age and gender effects.

## Figures and Tables

**Figure 1 sensors-21-03141-f001:**
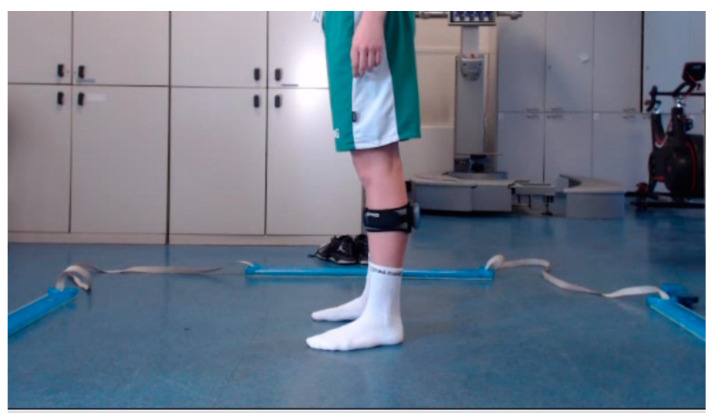
Subject equipped with the inertial sensor on the black stripe and placed within the area defined by the optoelectronic bars.

**Figure 2 sensors-21-03141-f002:**
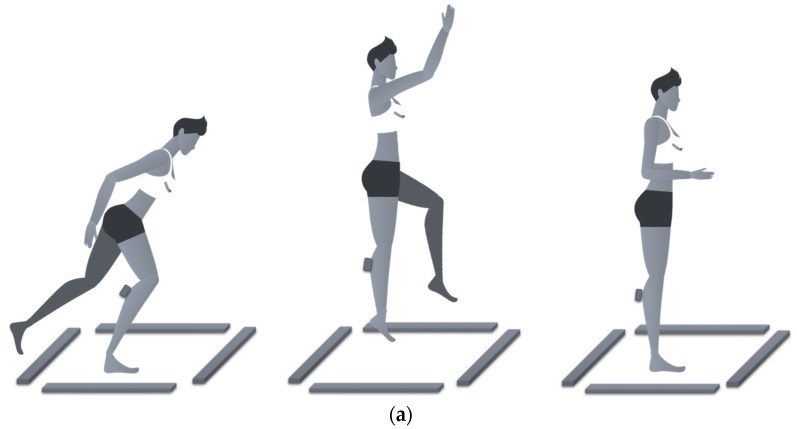
(**a**) Movement sequence related to the monopodalic countermovement jump and (**b**) movement sequence related to the single-leg squat. The four grey bars on the floor represent the optoelectronic bars, and the gray box on the subject’s shank represents the IMU.

**Figure 3 sensors-21-03141-f003:**
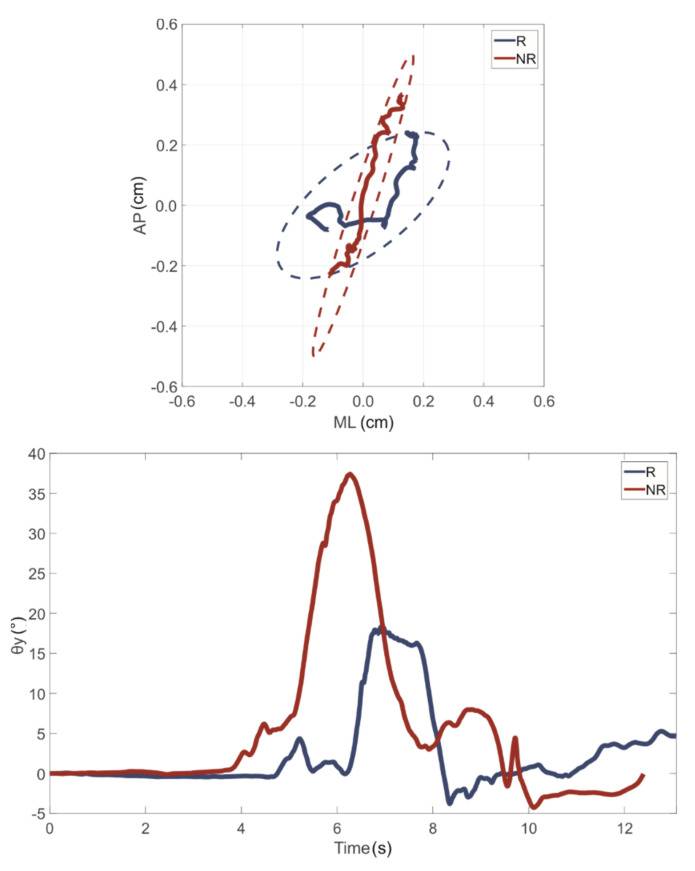
Example of (**top**) path length in xy plane during mCMJ and (**bottom**) angle waveform during SLS. Blue and red lines indicate results related to the subjects in the R and NR groups, respectively.

**Figure 4 sensors-21-03141-f004:**
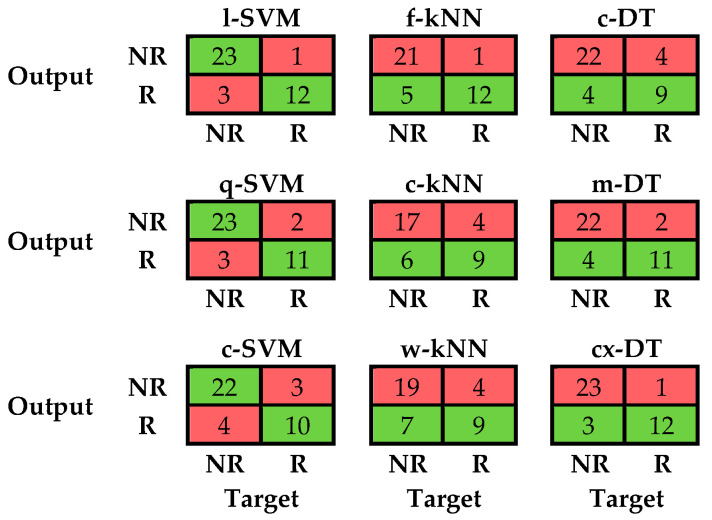
Confusion matrices for all the tested classifiers. NR and R stand for no risk and risk class.

**Figure 5 sensors-21-03141-f005:**
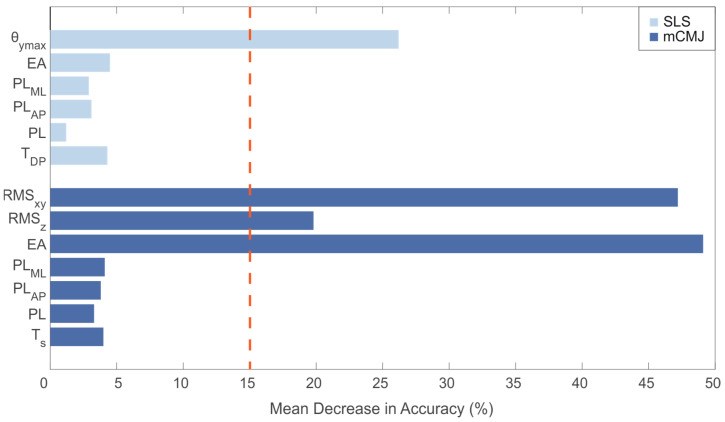
Mean decrease in accuracy (%) for all the used features. Dotted red line indicates the importance threshold of 15.0%.

**Table 1 sensors-21-03141-t001:** The 17 items for the evaluation of the LESS score.

Item	Possible Score
Knee flexion angle at initial contact > 30°	0 = yes 1 = no
2. Knee valgus at initial contact, knees over mid-foot	0 = yes 1 = no
3. Trunk flexion angle at contact	0 = trunk is flexed 1 = not flexed
4. Lateral trunk flexion at contact	0 = trunk is vertical 1 = not vertical
5. Ankle plantar flexion at contact	0 = toe to heel 1 = no
6. Foot position at initial contact, toes > 30° external rotation	0 = no 1 = yes
7. Foot position at initial contact, toes > 30° internal rotation	0 = no 1 = yes
8. Stance width at initial contact < shoulder width	0 = no 1 = yes
9. Stance width at initial contact > shoulder width	0 = no 1 = yes
10. Initial foot contact symmetric	0 = yes 1 = no
11. Knee flexion displacement before jumping > 45°	0 = yes 1 = no
12. Knee valgus displacement before jumping knee inside great toe	0 = no 1 = yes
13. Trunk flexion at maximal knee angle, trunk flexed more than at initial contact	0 = yes 1 = no
14. Hip flexion angle at initial contact, hips flexed	0 = yes 1 = no
15. Hip flexion at maximal knee angle, hips flexed more than at initial contact	0 = yes 1 = no
16. Joint displacement, sagittal plane	0 = soft 1 = average 2 = stiff
17. Overall impression	0 = excellent 1 = average 2 = poor

**Table 2 sensors-21-03141-t002:** Feature selection for both motion tasks.

Features		mCMJ	SLS
Leg stability	PL (cm)	✓	✓
	PLAP (cm)	✓	✓
	PLML (cm)	✓	✓
	EA (cm^2^)	✓	✓
Load absorption	RMSz (m/s^2^)	✓	
	RMSxy (m/s^2^)	✓	
Leg mobility	θ_ymax_ (°)		✓
Time parameters	Ts (s)	✓	
	T_DP_ (s)		✓

**Table 3 sensors-21-03141-t003:** List of tested machine-learning algorithms and acronyms.

Geometric		Binary
SVM	kNN	DT
Linear (l-SVM)	Fine (f-kNN)	Coarse(c-DT)
Quadratic (q-SVM)	Cosine (c-kNN)	Medium (m-DT)
Cubic (c-SVM)	Weighted (w-kNN)	Complex (cx-DT)

**Table 4 sensors-21-03141-t004:** Mean (SD) of the computed features for the R and NR group.

Task	Features	R	NR
mCMJ	TS (s)	0.21 (0.04)	0.33 (0.06)
	PL (cm)	2.7 (0.9)	2.9 (0.8)
	PL_AP_ (cm)	1.5 (0.5)	1.6 (0.3)
	PL_ML_ (cm)	2.8 (0.2)	1.1 (0.1)
	EA (cm^2^)	1.5 (0.2)	0.7 (0.2)
	RMS_z_ (m/s^2^)	247.4 (85.6)	95.6 (19.4)
	RMS_xy_ (m/s^2^)	169.4 (26.7)	70.1 (14.9)
SLS	T_DP_ (s)	3.22 (0.95)	3.90 (1.23)
	PL (cm)	0.8 (0.1)	0.8 (0.3)
	PL_AP_ (cm)	0.6 (0.1)	0.6 (0.2)
	PL_ML_ (cm)	0.4 (0.1)	0.5 (0.2)
	EA (cm^2^)	0.5 (0.0)	0.7 (0.0)
	θ_ymax_ (°)	16.8 (2.7)	26.0 (5.3)

**Table 5 sensors-21-03141-t005:** Accuracy (*A*), F1-score and goodness index (*G*) achieved by the nine tested classifiers.

	*A*	F1-Score	*G*
l-SVM	0.95	0.96	0.08
q-SVM	0.87	0.87	0.19
c-SVM	0.82	0.81	0.28
f-kNN	0.85	0.86	0.21
c-kNN	0.67	0.71	0.46
w-kNN	0.74	0.75	0.35
c-DT	0.79	0.79	0.34
m-DT	0.85	0.85	0.22
cx-DT	0.90	0.90	0.14

**Table 6 sensors-21-03141-t006:** Accuracy (*A*), F1-score and goodness index (*G*) achieved by the nine tested classifiers.

	r	*p*-Value
EA	0.88	0.01
RMS_z_	0.25	0.12
RMS_xy_	0.59	0.04
θ_ymax_	0.60	0.03

## Data Availability

For further details on data, please refer to the corresponding author.

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
