# Peer review of "A Machine-Learning Approach to Measure the Anterior Cruciate Ligament Injury Risk in Female Basketball Players"

_sensors, 2021, doi:10.3390/s21093141_

Round 1
Reviewer 1 Report
The early prediction of anterior cruciate ligament (ACL) injury still represents an open question. Within this aim, we proposed an innovative methodology based on machine-learning algorithms to discriminate against athletes at higher risk of ACL rupture. Thirty-nine female basketball players were enrolled in the experimental protocol, consisting of single-leg jump and squat tasks. Leg stability, leg mobility, and capability to absorb the load after jump landing were evaluated through inertial sensors and optoelectronic bars. The risk level of athletes was computed by the Landing Error Score System (LESS), used as a reference for the classification process. A comparative analysis among nine classifiers was performed by assessing the accuracy, F1-score and goodness. Five out of nine examined classifiers reached optimum performance, with linear support vector machine achieving the best one. The topic seems interesting, I have the following concerns to enhance the quality of the work.
- Your title is too long and should be reduced. A good title should not exceed 10 words.
The title should be clear and informative and should reflect the aim and approach of the work.
Recommendations for titles:
- Fewest possible words that describe the contents of the paper.
- Avoid waste words like "Studies on", or "Investigations on”, “effects of”, “comparison of”, or “a case of”
- Use specific terms rather than general
- Watch your word order and syntax
- Avoid abbreviations and jargon
- Authors should revise the abstract and accuracy should be added at end of the abstract.
- Why support vector machine implemented with linear kernel allowed achieving the best performance?
- I am not satisfied with the introduction of the paper. Introduction should provide a clearer view of the paper, as paper belong to computer-aided diagnoses so authors should think about it and should add few CAD papers in the introduction and related work section
.Claudino, João Gustavo, Daniel de Oliveira Capanema, Thiago Vieira de Souza, Julio Cerca Serrão, Adriano C. Machado Pereira, and George P. Nassis. "Current approaches to the use of artificial intelligence for injury risk assessment and performance prediction in team sports: a systematic review." Sports medicine-open 5, no. 1 (2019): 1-12. Khan, M. A., & Kim, Y. (2021). Cardiac arrhythmia disease classification using LSTM deep learning approach. CMC-COMPUTERS MATERIALS & CONTINUA, 67(1), 427-443.Ruiz-Pérez, Iñaki, Alejandro López-Valenciano, Sergio Hernández-Sánchez, José M. Puerta-Callejón, Mark De Ste Croix, Pilar Sainz de Baranda, and Francisco Ayala. "A field-based approach to determine soft tissue injury risk in elite futsal using novel machine learning techniques." Frontiers in psychology 12 (2021): 195. Yassin, Nisreen IR, Shaimaa Omran, Enas MF El Houby, and Hemat Allam. "Machine learning techniques for breast cancer computer-aided diagnosis using different image modalities: A systematic review." Computer methods and programs in biomedicine 156 (2018): 25-45.
- The research problem/ requirement is not elaborated properly.
- In the proposed method, the authors need to provide a solid scientific reason why the traditional feature selection methods are not enough to deal with the problem? why the proposed method provides high accuracy??
- The Authors need to explain how to handle class imbalance. It must be added to the proposed method.
- 7) Authors need to re-write the Abstract in a more meaningful way example (Problem definition=> How existing methods are lacking => proposed solution => Outcome
- All equations should be assigned numbers. And align with the text.
- All figures should be redrawn with high resolutions and different colors.
- The authors should give all experiment parameters in 2.2. Experimental setup, still few experiments paraments are missing??
8) Conclusion and Future work must be updated.
Reviewer 2 Report
This manuscript entitled “An innovative methodology to assess ACL injury risk in basketball players through machine-learning approach” aimed to propose an innovative methodology based on machine-learning algorithms to discriminate athletes at higher risk of ACL rupture. Although this study was read with interest, revisions are required to bring this up to a publishable standard. Some suggestions are listed below.
Specific suggestions
- Title, since only female basketball players were involved in this study, it is suggested that the title should be changed to “An innovative methodology to assess ACL injury risk in female basketball players through machine-learning approach”.
- Introduction, “In addition, ACL injury has the main incidence in basketball players together with the ankle sprains; for example, 16% of female athletes may incur an ACL injury during their career.”, it would be more logical if it was stated in this way, “In addition, ACL injury has the main incidence in female basketball players together with the ankle sprains; for example, 16% of female athletes may incur an ACL injury during their career.”.
- “Wearable solutions have been proposed in order to perform experimental tests also in playing field, overcoming the intrinsic limitation of the optoelectronic systems”, more details should be presented.
- The authors should develop the logical connection between the diagnosis and the prediction of the ACL injury.
- “Although it is evident the promising usefulness of the artificial intelligence for providing automatic information to both diagnose and predict ACL injury, the obtained results are not completely satisfactory to propose the use of such methodologies in daily routine.”, explain how the methodology used in this study would meet the daily routine requirements.
- The Methods would be demonstrated more clearly if the author could provide some figures of the experimental setup.
- Line 180 and Line 179 were completely the same.
- It is suggested that the Results should be separated from the Discussion.
- It is suggested that the Discussion should be rewritten since there was little comparison with previous studies, or any logical speculation based on the findings. Please focus on the main findings, explain the practical significance of the main findings, and give some logical and practical suggestions. Some recently studies could be referenced:
The Cutting Movement Assessment Score (CMAS) Qualitative Screening Tool: Application to Mitigate Anterior Cruciate Ligament Injury Risk during Cutting. Biomechanics 2021, 1, 83-101. https://doi.org/10.3390/biomechanics1010007
Single-Leg Landings Following a Volleyball Spike May Increase the Risk of Anterior Cruciate Ligament Injury More Than Landing on Both-Legs. Appl. Sci. 2021, 11, 130. https://doi.org/10.3390/app11010130
Effects of Eccentric Exercise on Skeletal Muscle Injury: From An Ultrastructure Aspect: A Review. Physical Activity and Health, 5(1), 15–20. DOI: http://doi.org/10.5334/paah.67
Rehabilitation and Return to Sport Assessment after Anterior Cruciate Ligament Injury: Quantifying Joint Kinematics during Complex High-Speed Tasks through Wearable Sensors. Sensors 2021, 21, 2331. https://doi.org/10.3390/s21072331
- Several simple examination maneuvers could be used to assess an athlete's control of knee motion. These include the single-leg squat and vertical drop jump tests. While useful assessment tools when performed by experienced clinicians, these examination maneuvers cannot predict ACL injury risk directly. Studies of such screening tests show considerable overlap between the results obtained from athletes who sustain an ACL injury and those who do not, and so no clear cutoff value can be established. The authors should explain whether their work could fill this gap.
- Any limitations of this study? Any future directions? Please specify.
- The Conclusion should also be further strengthened based on the findings of this study.
Round 2
Reviewer 1 Report
The authors did excellent work and resolve all my previous queries very well, this paper looks very interesting for the readers so now I agree to accept this paper in the present form for publication.